# Proximity to Neighborhood Services and Property Values in Urban Area: An Evaluation through the Hedonic Pricing Model

Asad Aziz [1,*], Muhammad Mushahid Anwar [1], Hazem Ghassan Abdo [2,3,4], Hussein Almohamad [5], Ahmed Abdullah Al Dughairi [5] and Motrih Al-Mutiry [6]

1   Department of Geography, University of Gujrat, Hafiz Hayat Campus, Gujrat 50700, Punjab, Pakistan
2   Geography Department, Faculty of Arts and Humanities, Tartous University, Tartous P.O. Box 2147, Syria
3   Geography Department, Faculty of Arts and Humanities, Damascus University, Damascus P.O. Box 30621, Syria
4   Geography Department, Faculty of Arts and Humanities, Tishreen University, Lattakia P.O. Box 30621, Syria
5   Department of Geography, College of Arabic Language and Social Studies, Qassim University, Buraydah 51452, Saudi Arabia
6   Department of Geography, College of Arts, Princess Nourah Bint Abdulrahman University, Riyadh 11671, Saudi Arabia
*   Correspondence: asadaziz@uog.edu.pk

**Abstract:** Neighborhood services, property attributes, and their associated amenities have positive impacts on land and property values. This impact is estimated by the hedonic pricing model, which is considered an effective method used in previous studies for such evaluations. The study uses Geographical Information Science by digitizing the point of interest in the study area for spatial modeling of data collection points and multi-linear regression as a statistical analysis of hedonic measurements. The hedonic measurements include the data of structural, locational, environmental, and community attributes of a property at a given time and space at a walkable distance from the neighborhood for measuring proximity. The results of the study are represented through the summary of the regression model, which expresses the impact of every individual variable on the entire value of the property, and the appropriateness of the results is shown by values R, $R^2$, and adjusted $R^2$. The result of the study concluded that property characteristics are varied from location to location, and that is why it is difficult to measure the exact market values, particularly in areas that lack urban planning and heterogeneous data. Research on such burning issues is essential for sustainable urban development.

**Keywords:** environment; hedonic model; land values; proximity; regression analysis; walkability

## 1. Introduction

In the urban world, the surrounding facilities have influential impacts on the property. A house or property at an accessible location is a basic unit for shelter in a settlement. Residential communities are the elementary living units in cities [1]. These units of living require different services and amenities, and one of them is neighborhood characteristics, which play an important role in defining the housing/property economy in a residential community, because they increase the competition between the demand for a property and the relative demand for a place [2]. The property values are significantly related to surrounding support facilities and community locations [1]. This study aims to provide the impacts of surrounding facilities on land and property values in an urban area of Gujrat by evaluating them through hedonic modeling because, in recent decades dramatically, property values have increased in the world; therefore, their assessment is needed for urban planning and management [3]. Economic fundamentals of property, land, and houses are recognized as determining factors, but the relationship between land and property prices is still disputed [4]. Effective urban planning requires known buyers about different amenities

and neighborhood characteristics [5]. There is a dire need to understand these dynamics in the housing market in an urban world.

Several research studies demonstrate a positive and strong relationship between property values and neighborhood characteristics [6]. However, the impact of this relationship between neighborhood features and accessibility on property values has not been well examined in the literature, particularly in the developing world [7]. This study also attempts to assess how the surrounding facilities (environmental, neighborhood, locational, and economic attributes) impact the property and land values in an urban area for the housing and property market because, in this area, no previous research has been done on this topic; however, very few studies have been found in the country for other locations [8]. It has been found that such studies can be considered for a land-allocation plan which provides the baseline for accessibility measurement. This accessibility measure provides a platform for regular walking platforms in a built environment [9].

This study takes into account the neighborhood properties comprising the location of the house/property, presence of a park near a house, proximity (walking distance) to major roads, hospitals, market area, and related facilities in one of the most congested urban areas of Gujrat, Pakistan [10]. Surrounding facilities have positive impacts on property value, and this association is well-acknowledged in past data [11]. The suitability of an area for the study is also a crucial issue during the concept development phases; however, the selected area is the most suitable, where all considered variables for hedonic measurements are applicable. The result of the study provides in-depth theoretical tools to urban geographers and concerned institutions for future planning, land allocation, and property sales and purchase decisions.

## 2. Review of Literature

A critical element for sustainable land use is a sustainable urban land-use policy [12]. The major change in land-use around the world is the expansion of the built-up environment and urbanization [13,14]. For achieving sustainable urbanization, it is critical to formulate policies and understand the spatiotemporal heterogeneity of the location and land use [15], where urbanization can be defined as the conversion of rural land into an urban area or its further development, which is growing rapidly [16]. Accessibility to surrounding facilities is a basic key measure for effective urban planning [17], and it also contributes to defining property values [18]. For policy making and urban planning, the impact of accessibility is valuable [19]. To evaluate the effectiveness of the policy-making process for an urban area, it is critical to evaluate this strategy in the context of sustainable urban development [12]. Association between neighborhood services and property values is an important aspect of the decision-making process in management strategies [20]. This association can be divided into other variables, which were discussed under the umbrella of structural, neighborhood/environmental, community, and locational attributes as considered in the hedonic model, for the formulation of the study.

### 2.1. Historical Background of the Hedonic Model

The word 'hedonic', meaning 'pleasure' has a Greek origin [21]; thus, hedonic pricing means 'pay for pleasure'. This model was first introduced by Rosen in 1974, who used the hedonic coefficients to understand the marginal willingness to pay for goodness and neighborhood amenities [22]. It means more neighborhood services, more relative goodness, and more relative values. The hedonic pricing model is one of the best models to measure the relative values of property in economic terms [23]. This is the leading model for the valuation of environmental services [24]. Despite its long history, this method is considered an active research tool for property assessment around the world [25]. Basic assumptions of the hedonic model include more attributes of a property and more demand, price, and sales ratio. The method has proved that access to public and private services, public amenities, as well as structural attributes shape the real estate market price of a house/property [26]. These public amenities include public places such as schools, banks, hospitals, parks, and

worship places, boosting property values [26]. A property that has more characteristics will have more utilities, and these utilities will increase the sales value of the property [27].

Description of the hedonic model: The basic assumption of this model is the evaluation of the impacts of neighborhood services on property values [28]. This model is considered a straightforward method and uncontroversial to apply; it lies in actual market values and data that can be collected directly from the field because the hedonic pricing model is used to estimate the total value of a property by the evaluation of local environmental and ecosystem services. The literature found that the traveling cost method is also used for the estimation of the influences of urban parks on the values of the property, but this method is valid if only traveling cost is investigated [29]. The hedonic pricing model measures the potential by dividing the total cost of property into separate costs for each characteristic or variable [30]. In this study, each variable is divided into sub-components such as the number of bedrooms, and bathrooms, the total area of the house/property, and proximity (walkability) to neighborhood services such as parks, markets, etc. The value of all these services defines the total cost of the property, and this method evaluates the willingness to pay for the attributes [31]. It has been observed that consumers make decisions not by a single character but by the number of characteristics that are counted to make decisions [32].

*2.2. Neighborhood Characteristics*

The associated characteristics such as social, environmental, or neighborhood will increase as well as decrease property values for those who live nearby and use them directly, for exercise, recreation, walking on green belts, similarly for ownership, societal events, traditions, schooling, and other usages [33]. While decrease in a sense, when these associated characteristics have negative effects, such as air pollution and traffic noise in rapidly growing cities on housing/property marketing, that is referred to as hedonic homogeneity [34]. These negative effects are auto-correlated in the sense that a house/property adjacent to a road has high accessibility, with the negative effect of traffic noise [34]. The local people directly attached to attributes will be affected. However, the disparity in prices is an unpredictable phenomenon temporally and spatially but will be considered when deciding on investment and purchasing a new property in the future.

Urban growth changes rapidly in every sphere of life in an area [5]. Considerably, in the least developing countries, urban planning has been done without the participation of the local community [35]. For policy making, urban planners need to check the influence of quality improvements on land values [36]; therefore, assessment and measurement are needed for financial and economic development. Finally, the outcomes of this study will provide more comprehensive knowledge that enhances the effectiveness of urban planning and policy making.

**3. Materials and Methods**

The fundamental principle of the hedonic model is the assessment of property values by determining the set of its characteristics [37]. These characteristics, in this case, study, are environmental services (E), community attributes (C), locational characteristics (L), and structural elements (S). Previous studies used different methods, including contingent valuation methods and hedonic wages methods, to estimate these measurements on property values; among all of these, the hedonic pricing method is the best found in the literature [38,39]. The previously described methods were found unsuitable for application due to the non-availability of data, lack of research methods for empirical results, and their fixed parameters that can not be applicable in some areas [40]. However, the hedonic model is designed based on a traditional measure, in the way of the proximity of spatial analysis, making it fixable to apply [10].

Following Figure 1 shows the framework of the whole methodology. In this formulation, a pilot survey was done to check the suitability of the area for the model. Furthermore, to check which variables have an influential impact on property values. This step is the

formulation of the topic and geographical analysis, which includes the mapping of the study area, mapping of data collection points (represented in yellow and purple color in the maps), and their distance from neighborhood services in terms of spatial analysis. The primary data related to hedonic variables were collected through a questionnaire Supplementary Material, and a regression model was applied to check the influence of each variable on the total property terms as data analysis, the hedonic model uses statistical methods (multi-linear regression) to estimate the price/values of property [41]. After that, results were obtained by running the regression analysis. Around the world, this model has been implemented by several researchers and professionals for property value estimation [42]. The model estimate and evaluation of the potential structure, environmental, and neighborhood attributes on property values [39]. According to area characteristics, the method is best suited due to flexibility and found a perfect candidate for detecting the proximity to neighborhood services and property values assessment through GIS and statistical measurements, as proved by [43].

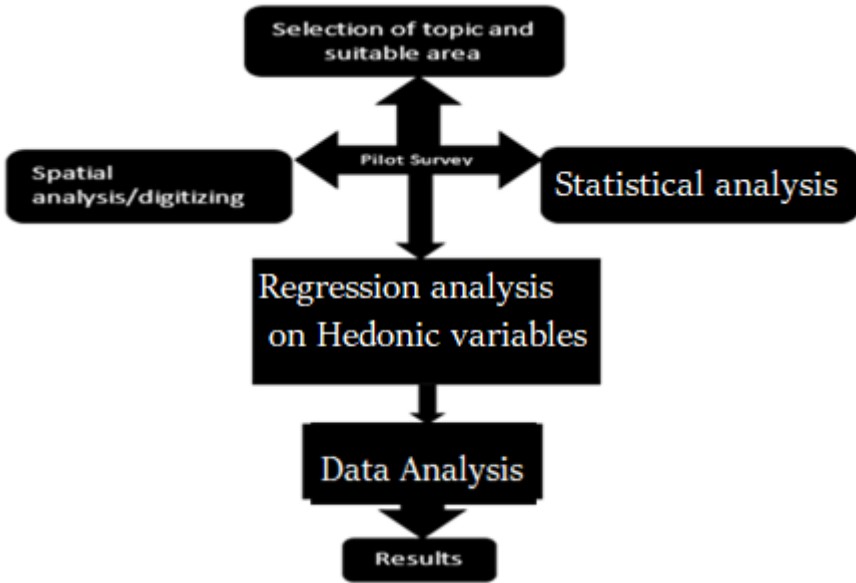

**Figure 1.** Framework of methodology.

### 3.1. Study Area

This study was done in a dense area of the population in an urban center named Children Park Town, Gujrat, which provides a suitable platform for the assessment of property values through the hedonic model. The area under examination contains a large portion of the settlement, a park, a dense network of roads and streets, which are counted for accessibility, and a market.

The Children Park Town is located in Gujrat with a total population of '1860 inhabitants' in the core of the city according to the recent census in 2017 [44]. All proposed variables for the hedonic measurement are best fitted in the area due to its location and associated neighborhood attributes. As property values are varying phenomena, the study only considers the previous five years' records from 2018 to 2022, respectively.

### 3.2. Methods

In this case study, the property value is a dependent variable in the regression model, including the price of land, and is equal to the sum of all costs and associated characteristics. These associated characteristics include the structural, environmental, neighborhood, and locational values taken as independent variables, and their link with the Geographical Information System (GIS) makes them a spatial hedonic model [45]. Equation (1) below shows the potential of considered variables in hedonic terms; Adopted: [46]:

$$P_{RS} = (S, N\ L, C) \tag{1}$$

where $P_{RS}$ (property value) is a dependent variable, depending on values of structural properties (S), neighborhood characteristics (N), community attributes (C), and locational attributes (L) that are independent and heterogeneous.

The results of the study are checked through the coefficient of variations, significant value ($p < 0.05$) for each variable. Each neighborhood's characteristics have a unique value to define the total cost of the property, and little changes in them will affect the total cost according to the hedonic pricing model. The study shows that the impact of proximity provides a base for a sustainable city, which is also defined by [47], and this impact is allied with economic, social, financial, and psychological issues [47]. Because walkability shapes residential behaviors and the social environment, worldwide land-use managers tend to focus on these impacts [48]. The methods concluded that understanding these relationships between community deprivation and walkability is an appropriate platform for land-use planning [48].

The mathematical formulation for the hedonic variables, Adopted: [46]

$$\sum P_{RS} = \beta_c\ X_c + \beta_L\ X_L\ ^+ \beta s X s + \beta_n\ X_n + \varepsilon \tag{2}$$

where $P_{RS}$ = hedonic value (total price of the property); $\sum$ = sum of all considered variables; $\varepsilon$ = error term.

$X_{L\ =}$ Represent locational attributes. $Xs$ = Represent structural attributes.

$X_c$ = Represent Community attributes. $X_n$ = Represent neighborhood attributes.

The independent variables are heterogeneous, vary from positive to negative potential in terms of value on total price, and differ from location to location and property to property. All of these major four attributes are further divided into thirteen different variables according to hedonic calculations. The economic estimation of these variables was calculated using an equation in the multi-linear regression statistical model [49] that used the Statistical Product for Social Sciences (SPSS). Meanwhile, the proximity measurements were estimated through GIS by using Google Earth Pro and Google Maps to measure the area and distance from the hospital, market, major road, and structure of the house. GIS mapping provides a spatial view of the area under examination and data-collection points along roads and street networks in the study area in Figure 2.

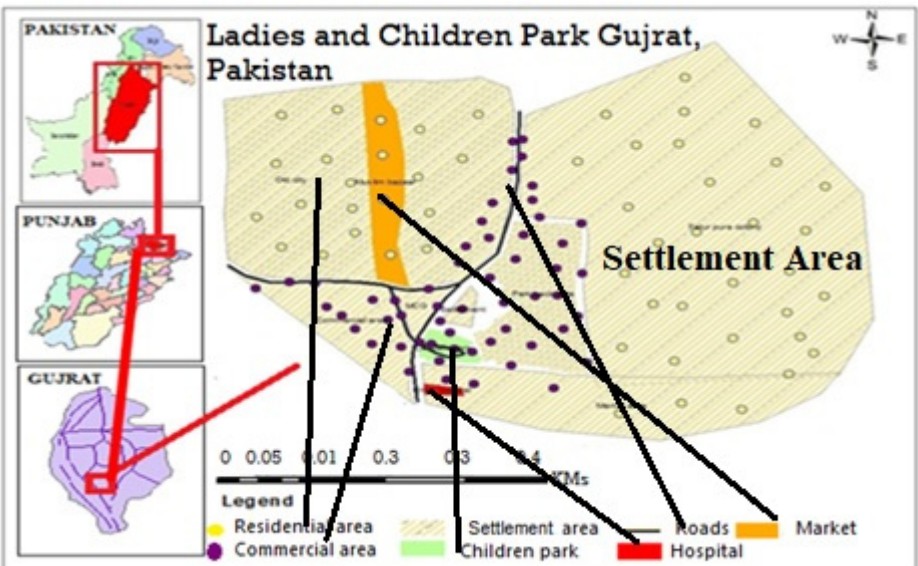

**Figure 2.** Map of Study Area.

*3.3. Data Collection*

In this case study, the primary data were collected through a random sampling procedure from the study area. The questionnaire consists of two major portions: the qualitative part (related to the socioeconomic profile of interviewees, age, gender, job status, and marital status) and the quantitative part, including the location of the property, neighborhood services, stories in the building, number of bedrooms and bathrooms, and the social status of people in the surrounding, middle class or low class taken as social variables. In this formulation, the hedonic model variable portion has weightage as it contains the base of the research. The secondary data (the price of land in sq. ft) related to the property were collected from Land Record Authority Punjab, Urban Gazette, and public property dealers in the study area for the years 2018 to 2022. The decision was made based on numerical values obtained from the results. The assessment and evaluation were done by measuring the significant values of each variable.

**4. Results**

The impact of proximity to the neighborhood services on property values in human lives was found to be positive with mixed, measured values, and the results of the study also provide evidence in this regard. This evidence will certainly provide support for effective urban planning and management. The results of this evaluation and assessment are not only important for the urban administration but also for property consultants, local government, local community, administration sustainability, and sale/purchase decisions in housing/property marketing. In this study, the researcher is concerned with how the neighborhood's services determine the property values. The study shows dynamic results, and there are several reasons, such as people's preferences, the location of the property, the socioeconomic profile of the local population, and diverse characteristics.

The following Figure 3 shows the land values of the selected location by using purple dots (with high land values) and yellow dots (with low values due to a greater distance from the road). The study concluded that the greater the area of property, the greater the price, the larger the stories, and the higher the values, and this phenomenon is also proven by previous studies [50].

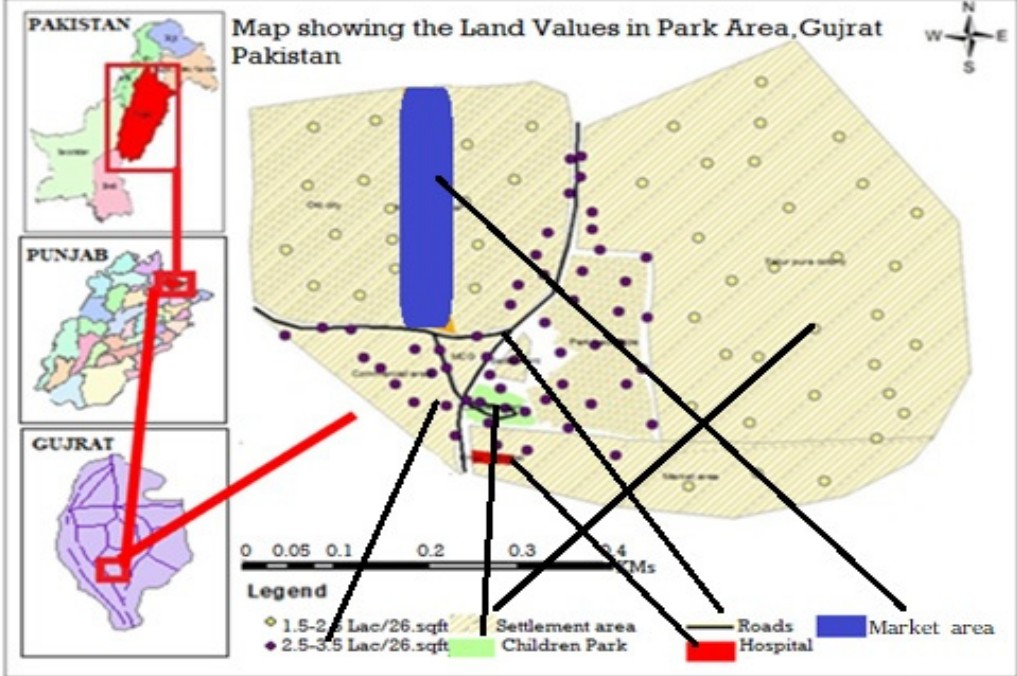

**Figure 3.** Map showing the land values and proximity.

After the resultant analysis above, Figure 3 expresses the land near the roadside having PKR 250,000 to 350,000 price per Marla (26 sq. meters), whereas the land and property located within the settlement have fewer prices in ranges from PKR[1] 150,000 to 250,000 per Marla (26 sq. meters). The land located in commercial areas has high prices as compared to residential areas due to wide road networking, marketing areas, and people's preference for commercial places and market environments for business.

Table 1 shows the regression model summary of the selected fourteen variables, where the value of the R coefficient of regression shows a positive correlation between dependent and independent variables, which means that the applied model is suitable for price estimation of selected hedonic variables. Likewise, $R^2$ shows the coefficient of determination having a value of 0.832, expressing a significant association, which means that thirteen independent variables considered in the study have an 83% influence on the dependent variable. Similarly, the adjusted R square value 0.806 is explanatory power.

**Table 1.** Regression summary for the selected variables.

| R | $R^2$ | Adjusted R Square | Std Error of the Estimation |
|---|---|---|---|
| 0.912 | 0.832 | 0.806 | 694,033.9125 |

Source: Author's calculations, 2023.

Table 2 illustrates the overall performance of the regression results by ANOVA analysis. According to regression suitability analysis, the values of significance should be less than the values of "F" [28]. In light of the above results, the value of significance is less than that of the "F" value. Therefore, the results of the regression model are positive and ultimately significant.

**Table 2.** ANOVA Analysis.

| Model | Sum of Squares | Df | Mean Square | F | Sig. |
|---|---|---|---|---|---|
| Regression statistic | $22.050 \times 10^{14}$ | 13 | $1.577 \times 10^{13}$ | 35.735 | 0.000 |
| Residual statistic | $41.42 \times 10^{13}$ | 86 | $4.817 \times 10^{11}$ | ***[2] | *** |
| Total | $63.47 \times 20^{17}$ | 99 | *** | *** | *** |

Source: Author's calculations, 2023.

The regression model has been applied at a 95% confidence level. Table 3 shows the coefficient of each independent variable individually 13 variables as represented in mean square, while the t-statistics is calculated using a ratio of coefficient to the standard error of variables. The confidence level is set at 95% with a 5% margin. The "*p*" value for eleven variables is more than 0.05, meaning that the variable is not affecting the total sale value of a property, as shown in Table 3.

The significant results (statically significant *p* values > 0.05) were obtained for locational variables, including house/property location, size of the property, number of stories, number of rooms/bathrooms, total covered area, and distance to markets due to people's preferences. Furthermore, locational variables, including access to roads, land nature in the surrounding of a house, and nature of communities, have negative results with negative coefficient values. The major cause for negative values is congested and narrow streets, the locality in a dense array of populations, a congested housing network, and heterogeneity in land use. The property variables show insignificant results. The population prefers to live away from hospitals, as people feel unpleasant about the hospital surrounding/medical environment. For community and neighborhood variables, including the social and job status of the population, the negative results were caused by urbanization pressure as the population of different areas settled here, which belongs to different social classes

and is involved in low-class business for survival. This effect creates heterogeneity in the socioeconomic status of the local population.

**Table 3.** Regression coefficients.

| Hedonic Variables | $\beta$-Beta Values of the Regression | Critical-T-Values | Significant-$p$-Value |
|---|---|---|---|
| Location of Property | −0.008 | −0.124 | 0.902 |
| Access to facilities | 0.071 | 1.023 | 0.309 |
| Size of property | 0.075 | 1.316 | 0.192 |
| Stories in building | 0.253 | 3.949 | 0.000 |
| Rooms in house | 0.324 | 3.578 | 0.001 |
| Bathrooms in house | 0.103 | 1.185 | 0.239 |
| Covered area by property | 0.199 | 4.121 | 0.000 |
| Land price in PKR | −0.025 | −0.271 | 0.787 |
| Land Nature (residential or commercial) | −0.034 | −0.471 | 0.639 |
| Distance to hospital | −0.134 | −1.570 | 0.120 |
| Distance to market | −0.048 | −0.750 | 0.455 |
| Nature of community in surrounding | −0.116 | −1.572 | 0.120 |
| Community jobs in the surrounding of a property | −0.123 | −1.817 | 0.073 |

*Result of Suitability Analysis*

The regression model suitability was checked through the regression line graph, as expressed in Figure 4.

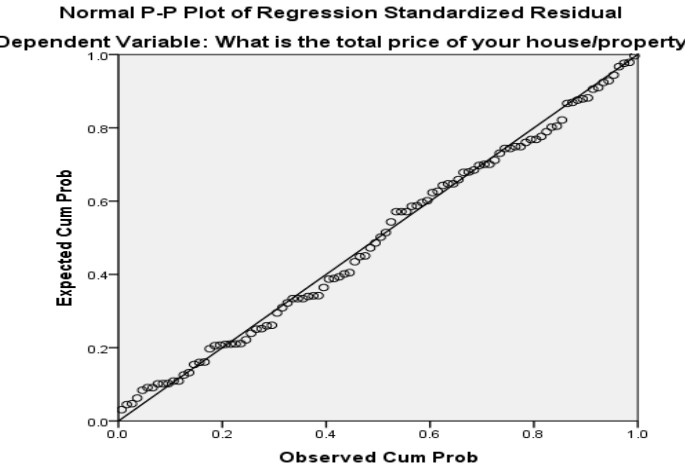

**Figure 4.** Scatter plot showing the regression line graph for applied study.

## 5. Discussion

The study found a positive correlation between land and property price due to proximity effects to the road and major street networks. Some people, particularly the educated, prefer the accessibility to the green area and parks near their homes in the dense urban region. A property near a road has a high value as compared to a property at a distance [51]. However, some previous studies show dynamic results, including people's preferences, their choice, the socioeconomic profile of the concerned population, and varying characteristics of the house and properties such as stories in a building and age [52]. A buyer, investor,

and renter will count all these characteristics during purchasing or selling a property as investigated during research [53]. This study attempts to apply this model to a new area in a developing country where hedonic variables are heterogeneous due to a lack of urban planning and provide valuable suggestions for urban planning.

The characteristics of land and property are varying phenomena temporally and spatially [54]. These measurements will determine the decision for the sale or purchase of a house or property and investments [55]. The evaluation of these measurements is accurately investigated by hedonic modeling, as proved by the study. Around the world, this model is implemented by several researchers and professionals for property valuation [42], with varied characteristics of houses or properties at multi-geographical scales [41]. However, the common functionality in all, the model estimates and evaluates the potential of structural, environmental, and neighborhood attributes on property values [39]. The major reason behind the application of the model is flexibility in variable selection, choice, and modification according to the heterogeneous nature of an area.

It has been observed that the choice of houses and property buying is a varying phenomenon, even from culture to culture, along with social, economic, and environmental factors [56]. However, among all the environmental factors are influential in deciding to buy a house or property. These factors include the presence of a green area near the house, walkable distance to the city center, community gardens, and other public facilities [57]. This study is the first attempt in this region to estimate the impacts of neighborhood services on property values. Therefore, it will provide a baseline to open the doors of further research on such topics. The recent trends of the local market in this area are more precise and consider the environmental factors [37], including air quality and open spaces in the housing network for playgrounds and parks as found [58]. Finally, very few studies on such topics have been found particularly in the least developing countries; in this perspective, however, this research provides a fundamental approach and also opens the door for further research.

## 6. Conclusions

The study found heterogeneous impacts of neighborhood services on property values due to different spatial and temporal effects. However, locational attributes are the influential factors determining the value and prices of a property. Most people purchase and prefer land in a porch area for housing and settlement, which is also proved by [37]. The choice of house and property is a varying phenomenon because of different social, economic, and environmental factors, which can be changed from region to region and country to country. It has been investigated that the rapid rate of urbanization and migration from rural to urban areas creates multiple challenges for city administration and fluctuation in land values along with housing choice. For urban development, it is very necessary to monitor this migration pattern to mitigate the challenges.

There is a strong need for an institution to register and keeps records related to property and housing profile, including location, owner information, housing characteristics, and age. The varying characteristics of houses and properties are also challenging tasks due to the lack of planning for the model applied. This is the reason the land allocation mechanism needs to be adopted for city administration and the sustainability of an area. It should be based upon the principles of land use regulations in an urban area, and the allotment of the land to hospitals, markets, and public places should be done according to urban land-use modeling. For proximity measures, the major roads should be connected with internal street networks with a proper mechanism to provide equal accessibility to the local population. The market area should be separate from the residential area to create homogeneity in land use. The educational institutions and government should adopt and focus the research on such crucial issues to promote good governance and society's well-being. The engagement of locals to understand their needs and demands and then design the urban landscape and housing market is needed. The stakeholder should encourage and engage the people to share their data for educational and research purposes for better

land-use planning and management, as suggested by [37]. This is the best way to achieve social well-being and geographical equality, and equity.

The major limitation involved in this research is the absence of data related to the market values of the property; often, people hesitate to share their property data for social/personal constraints. The absence of related data is an influential issue in hedonic research, as found in our study. However, the authors collect data from official sources such as Punjab Land Record Authority for study purposes. It has been noted that such types of studies if done in the least developing countries, the outcome will be useful for property planning and provide a baseline for developing strategies. This will help to accomplish the 11th UN goal (Sustainable cities and communities) for sustainable development under the umbrella of the United Nations Development Program (UNDP).

**Supplementary Materials:** The following supporting information can be downloaded at: https://www. mdpi.com/article/10.3390/land12040859/s1, Supplementary Material: Questionnaire the Impact Of Green Spaces on Sale's Property Values through the Hedonic Model.

**Author Contributions:** Conceptualization, A.A., M.M.A. and H.G.A.; methodology, A.A. and M.A.-M.; software, A.A., M.M.A., H.A. and M.A.-M.; validation, M.M.A. and A.A.; formal analysis, A.A.; investigation, A.A., A.A.A.D. and M.M.A.; resources, A.A. and M.M.A.; data correction, A.A., M.M.A. and H.A.; writing—original draft preparation, A.A., M.M.A. and H.G.A.; writing—review and editing, A.A., M.M.A., H.A., A.A.A.D. and M.A.-M.; visualization, A.A., M.M.A., H.A., A.A.A.D. and M.A.-M.; supervision, A.A., M.M.A., H.A., A.A.A.D. and M.A.-M.; project administration, A.A., M.M.A., H.A., A.A.A.D. and M.A.-M.; funding acquisition A.A., M.M.A., H.A., A.A.A.D. and M.A.-M. All authors have read and agreed to the published version of the manuscript.

**Funding:** This project was funded by Princess Nourah bint Abdulrahman University Research Supporting Project Number PNURSP2022R241, Princess Nourah bint Abdulrahman University, Riyadh, Saudi Arabia.

**Data Availability Statement:** Not applicable.

**Acknowledgments:** We thanks to our cross-ponding author Asad Aziz for provision of data from his M.Phil thesis for the write-up of this paper.

**Conflicts of Interest:** All authors declare no conflict at any stage of the study.

## Notes

[1]　Lac is equal to 0.1 million.

[2]　Values are not available in the Regression result ANOVA Table.

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
