# Peer review of "Proximity to Neighborhood Services and Property Values in Urban Area: An Evaluation through the Hedonic Pricing Model"

_land, doi:10.3390/land12040859_

Round 1

Reviewer 1 Report

The paper is suitable for the journal. The paper is about using geographic information systems to determine land value.

The abstract summarises well the paper.

The paper is generally good, but there are some things to correct:

Fig. 1 puts action plan in the same square as data analysis. This is wrong. An action plan is part of strategic planning which is not properly included.

The other figures are low res and distorted, please correct.

To be discussed is if not also proximity to green spaces should be included.

Author Response

Reviewer 1

Comments and Suggestions for Authors

The paper is suitable for the journal. The paper is about using geographic information systems to determine land value.

Thank you for your comments.

The abstract summarises well the paper.

Thank you for reading our paper

The paper is generally good, but there are some things to correct:

Fig. 1 puts action plan in the same square as data analysis. This is wrong. An action plan is part of strategic planning which is not properly included.

Dear reviewer, please note figure 1 is about the framework of methodology as cited in material and methods in first portion of this section. However, the sentence has been changed and action plan is replaced by suitable words.

The other figures are low res and distorted, please correct.

We try to make changes at best possible level, I hope it work

To be discussed is if not also proximity to green spaces should be included.

Alongside some people particularly educated, prefer the accessibility to green area, parks near to their home in dense urban region. ((((added in the text file))))

Reviewer 2 Report

  • A brief summary Theme of the paper is acute and well-formulated. Structure ao the paper is correct, methodology is adopted from other relevant research. Referencies are relevant, recent and relevant
  • General concept comments
    However, there are serious comments to the article and its authors.
  • Aim, goal, hypothesis of the research are not formulated. It is impossible to check significance of the results, because authors do not provide relevant info, e.g. there is no a number of records/cases analysed, questionnaire is not well-presented, Methodological inaccuracies: authors adopted methodology by Sisi Yan (2012), made some corrections  but did not explaine the changes. Paper is superficial, without an in-depth analysis of the researched processes and obtained results. Formulas and figures contain inaccuracies. There is no clear description of the variables and their measurement units. Therefore, the author's interpretation of the results raises doubts. The structure of the second section needs to be changed, because it is illogical and contains repetitions. The main thing: the article needs a scientific editing of the text, because there are many incorrect and/or unclear sentences. And the edition of the English language is needed
  • Specific comments 

· The basic assumption of this model  is not clearly formulated  (109-110)

·         There is not a description of the paper aim and tasks

·         Introduction is written as separated sentences, without linkers. There is not a clear vision of what, when and where is done in the research area. Some paragraphs could be moved to Section 2

·         Methodology: questionnaire is presented vaguely. It is not clear what authors consider as secondary data (192). Description of variables is not clear and precise (182-187). There is no clarity/sequence of the material presentation. Time span is too broad:  2016- 2020. And there is not information on recent trends on local markets. It is better to begin with 2.1. Study area and then 2.2 Methods and finally 2.3 Data collection. And remove some repetitions in lines 190-162 and 262-263. Mathematical Terms for the Hedonic Variables put in part 2.2. Methods. Clarification of independent variables should be submitted immediately after the presentation of the model (2). The independent variable is spelled differently in lines 229 and 231 as well as in lines 251 and 253. Text in lines 234-237 is not understandable. Several comments to model (2): The purpose of the sum signon thebeginning og thr formula  is unclear, because the summation index is not specified. Pls check [42] once more. In [42] natural logarithms are used. Here authors don-t explain their modification. Indexes differ in writing. Why are two "+" signs written in a row? Some signs "plus" are written as subscripts. Phrase “∑= Sum of all considered variables” (253-254) is not correct because you already used sign “+”. Authors write that “The independent variables are continuous”. It does not, e.g two bathrooms, three levels etc.

·         The last paragraph of Section 2 should be moved to Section 3. The same is relevant to lines 238-249.

·         Results: authors use term “The positive results” (329) but is not clear. May be statistically significant results? The results (238-248) are not in place

·         Figure 1: several names of the blocks are not correct: Hedonic analysis for statistical analysis – last three words are redounded; Identification of relevant approaches – approaches to what? Action plan in the end of the scheme is not clear, last block should correlate with the goal of the paper

·         Fig. 3 is like Fig 2. Figure 3 is named as “Map showing the land values and proximity” but land values and proximity are not indicated on the figure. And there is a typo: markAt.

·         Table 1. Caption of this table is “Table 1-Regression Model Results” but table presents only part of results. So this name is not precise. Critical F-value and a number of records analysed is needed

·         In Tables 1 and 2 not all columns have names.

·         Table 3: authors should indicate a number of records analysed, critical t-value and mark with asterisk all significant coefficients

·         Term ”traveling cost method” is not correct (114) and paper [27][1] does not apply it

·         125-127: Is not clear Why authors expect a decrease of property values when the associated characteristics like social, environmental, or neighborhood will increase. 160-163; 163-166

·         Line 263: if you write about proximity is it a walking distance?

·         Std Error of the Estimation is very high. Are you sure it should be squared? Pls  check it once more.

·         Model is multivariable thus sign of each variable is unknown. Not positive, as authors write in line 302. Even more. In table 3 there are significant coefficients with a negative sign.

·         Coefficients for house/property location and bathrooms are not significant – pls check text  (329)

·         Discussion about people’s preferences concerning proximity to road and location of a  property and its positive correlation  with a property price (414-415) is not clear, because relevant coefficient is negative and nonsignificant and authors did not explain how they coded the variable

·         For all variables authors should explain meaning, coding and units of measurement.

·         The paper need scientific corrections because there are a lot of incorrect statements from a scientific point of view, e.g.: “Rosen's in 1974 gives the results of hedonic coefficients”  94-95; not “on total property” but on total property value  or price; Model does not use methods (167), Scientists use them; “The evaluation of these questionnaires was done through the Multi Linear Regression Technique” – perhaps authors write about data analysis or evaluation of regression coefficients (190)

·         English:  phrases “These hedonic portion include the data” (186), “Here the meaning of total values means larger the area or structure the structure of 234 house more the price.” (234-235), “such types of types if done the outcome will be useful for property 432 planners” (432-433), lines 49, 59-61; 69-73, 81-82 etc.

[1] Numbers in parentheses mean the line number in the text of the article

Author Response

Reviewer 2

Comments and Suggestions for Authors

  • A brief summary Theme of the paper is acute and well-formulated. Structure of the paper is correct, methodology is adopted from other relevant research. References are relevant, recent and relevant
  • Thank for your comments
  • General concept comments
    However, there are serious comments to the article and its authors.
  • Aim, goal, hypothesis of the research are not formulated. It is impossible to check significance of the results, because authors do not provide relevant info, e.g. there is no a number of records/cases analysed, questionnaire is not well-presented, Methodological inaccuracies: authors adopted methodology by Sisi Yan (2012), made some corrections  but did not explaine the changes. Paper is superficial, without an in-depth analysis of the researched processes and obtained results. Formulas and figures contain inaccuracies. There is no clear description of the variables and their measurement units. Therefore, the author's interpretation of the results raises doubts. The structure of the second section needs to be changed, because it is illogical and contains repetitions. The main thing: the article needs a scientific editing of the text, because there are many incorrect and/or unclear sentences. And the edition of the English language is needed

Thank you very much for the comments, the whole of the study is now carefully investigated, and improvements has been made including the English language editing, structure of the sentence, by adding objectives, hypothesis and results.  The study also modified the sentences for scientific writing to make the concept more clear.

  • Specific comments 
  • The basic assumption of this model  is not clearly formulated  (109-110)

This study aim to provide the real picture of these surrounding facilities on land and property price and values in urban areas. Because, in recent decades dramatically, property values has increased in the world [3]. Economic fundamentals of property, land and houses are recognized as determining factors but the relationship among land and housing price is still disputed [4]. Effective urban planning requires the known buyers about different amenities and neighbourhood characteristics [5]. Anyhow, there is dire need exist to understand these dynamics in housing market in urban world.  (((( this sentence is add in the paper))))

  • There is not a description of the paper aim and tasks

We re-define the basic concept, hope it work

  • Introductionis written as separated sentences, without linkers. There is not a clear vision of what, when and where is done in the research area. Some paragraphs could be moved to Section 2

We try to makes changes by keeping these considerations, some paragraph are divided into subsections and last para in moved to section 2.

  • Methodology:questionnaire is presented vaguely.we add information abut the questionnaire  It is not clear what authors consider as secondary data (192). We add the information about secondary data. Description of variables is not clear and precise (182-187).we redefine this to make it clear  There is no clarity/sequence of the material presentation.we improve the methods sections. Time span is too broad:  2016- 2020, And there is not information on recent trends on local markets.Information about recent time is now added. It is better to begin with 2.1. Study area and then 2.2 Methods and finally 2.3 it is changed accordingly.  Data collection. And remove some repetitions in lines 190-162 and 262-263. Its now replaced. Mathematical Terms for the Hedonic Variables put in part 2.2. Methods. It is write in this section in equation 1 and 2. Clarification of independent variables should be submitted immediately after the presentation of the model (2). The independent variable is spelled differently in lines 229 and 231 as well as in lines 251 and 253. Text in lines 234-237 is not understandable. Several comments to model (2): The purpose of the sum signon thebeginning og thr formula  is unclear, because the summation index is not specified. All of these comments are considered while revisions and changing has been done. Pls check [42] once more. In [42] natural logarithms are used.The previous [42] and now [43] is a web page of govt of Pakistan. Here authors don-t explain their modification. Indexes differ in writing. Why are two "+" signs written in a row? Some signs "plus" are written as subscripts. Phrase “∑= Sum of all considered variables” (253-254) is not correct because you already used sign “+”. Authors write that “The independent variables are continuous”. It does not, e.g two bathrooms, three levels etc its was our mistake its bed rooms now its changed and correct in order.. Sigma is the part of sum of all independent variables and part of regression equation., that’s why we used it in the start of the equation and plus sign for all variables individually.
  • The last paragraph of Section 2 should be moved to Section 3. The same is relevant to lines 238-249.

It’s moved to section 3

  • Results:authors use term “The positive results” (329) but is not clear. May be statistically significant results? The results (238-248) are not in place

The positive results (statically significant p values> 0.05) were obtained for house/property location, size of property, stories, rooms/bathrooms, total covered area and distance to markets (Table-3

  • Figure1: several names of the blocks are not correct: Hedonic analysis for statistical analysis – last three words are redounded; Identification of relevant approaches – approaches to what? Action plan in the end of the scheme is not clear, last block should correlate with the goal of the paper

We rewrite these sentences, please check.

  • Fig. 3 is like Fig 2.Figure 3 is named as “Map showing the land values and proximity” but land values and proximity are not indicated on the figure. And there is a typo: markAt.

The following figure-2 and 3 shows the land values and also representing the proximity effect on property values by using purple (with high land values, and yellow dots (with low values) due to more distance from the road  .Typo is correct now. Changes has been made in the paper also.

  • Table 1.Caption of this table is “Table 1-Regression Model Results” but table presents only part of results. So this name is not precise. Critical F-value and a number of records analysed is needed.

Title for table 1 is changed, total 14 variables were analyzed and F value is provided in table 2

  • In Tables 1 and 2not all columns have names.

Changes has been made and highlighted yellow at the top in both tables. However, these tables are automatically get from SPSS software. 

  • Table 3:authors should indicate a number of records analysed, critical t-value and mark with aste risk all significant coefficients
  • Term ”traveling cost method” is not correct (114) and paper [27][1] does not apply it

we recheck the reference, and update with correct order.

  • 125-127: Is not clear Why authors expect a decrease of property values when the associated characteristics like social, environmental, or neighborhood will increase. 160-163; 163-166

It is due to public preference and heterogeneous nature of data varied from location to location.

  • Line 263: if you write about proximity is it a walking distance?

Yes it is walking distance, meaning has be added

  • Std Error of the Estimation is very high. Are you sure it should be squared? Pls  check it once more..

It is now corrected. Its not squared

  • Model is multivariable thus sign of each variable is unknown. Not positive, as authors write in line 302. Even more. In table 3 there are significant coefficients with a negative sign.

We mentioned the areas is heterogeneous in nature, some people prefer structural variables some prefer locational, which will effect the coefficient values, it is also mentioned in paper file

  • Coefficients for house/property location and bathrooms are not significant – pls check text  (329)

The whole sentence is rewrite now

  • Discussion about people’s preferences concerning proximity to road and location of a  property and its positive correlation  with a property price(414-415) is not clear, because relevant coefficient is negative and nonsignificant and authors did not explain how they coded the variable

The discussion is changed, we rewrite this section with property structure.

  • For all variablesauthors should explain meaning, coding and units of measurement.

All variables in table 3 are explained for better understand. These questions were open ended, there is no coding.

  • The paper need scientific corrections because there are a lot of incorrect statements from a scientific point of view, e.g.: “Rosen's in 1974 gives the results of hedonic coefficients”94-95; not “on total property” but on total property value  or price; Model does not use methods (167), Scientists use them; “The evaluation of these questionnaires was done through the Multi Linear Regression Technique” – perhaps authors write about data analysis or evaluation of regression coefficients (190)

Many thanks for all of these corrections, after carefully consideration all of these corrections has been done. We are very grateful to the reviewer for making such amazing comment.

  • English: phrases “These hedonic portion include the data” (186), “Here the meaning of total values means larger the area or structure the structure of 234 house more the price.” (234-235), “such types of types(Studies) if done the outcome will be useful for property 432 planners” (432-433), lines 49, 59-61; 69-73, 81-82 etc.

All these corrections also has been done in the paper file.

[1] Numbers in parentheses mean the line number in the text of the article

Alright I understand

Reviewer 3 Report

- The introduction is too long, two and a half pages. You can summarize the introduction and include a separate section for the literature review before going into the methods.

- The last paragraph of the introduction can highlight the objectives/questions and the last paragraph of the literature review can highlight the gap in the knowledge. 

- The literature (which is currently mainly part of the introduction) can be enhanced and sub sectioned according to different themes. 

- Figures 2 and 3 are important but needs to be redesigned for better and clearer readability. 

- The discussion section could benefit from re-writing. The purpose of the discussion is to explain the connection between the results of the study and existing knowledge. To fill the gap. Currently, the discussion does not tie existing knowledge with solid findings. Perhaps a round of re-writing could help emphasis more on the gaps in the knowledge and the findings of the study. 

Author Response

Reviewers 3

Comments and Suggestions for Authors

- The introduction is too long, two and a half pages. You can summarize the introduction and include a separate section for the literature review before going into the methods.

Thanks for your comments; the introduction section is divide into two section/subsections

- The last paragraph of the introduction can highlight the objectives/questions and the last paragraph of the literature review can highlight the gap in the knowledge. 

It is re-positioned according to the directions/suggestion of one of our reviewers.

- The literature (which is currently mainly part of the introduction) can be enhanced and sub sectioned according to different themes. 

We makes two sections/subsection including (history of the hedonic model and its basic assumptions). Hopes it is correct in order now.

- Figures 2 and 3 are important but needs to be redesigned for better and clearer readability. 

We try at level best, hope it will work

- The discussion section could benefit from re-writing. The purpose of the discussion is to explain the connection between the results of the study and existing knowledge. To fill the gap. Currently, the discussion does not tie existing knowledge with solid findings. Perhaps a round of re-writing could help emphasis more on the gaps in the knowledge and the findings of the study. 

Thank you for your comments; we rewrite the discussion and conclusion section once again with proper structure.

Reviewer 4 Report

Proximity to neighbourhood services and property values in urban area: An evaluation through Hedonic Pricing Model

Comments and Suggestions for Editor and Authors

The use of hedonic price models is an unavoidable topic of research on prices in the real estate market. Although there are international studies on the use of hedonic models, the authors refer that in their country it is a pioneering work, so it will be an important work for that region of the globe. In addition, it is very important for the academic community to develop research on a current topic, such as this case. The work is well-structured. The abstract, introduction, and literature review are written carefully and balanced. The aim of the work could (should) be mentioned in the introduction and at the beginning of the conclusion, thus giving "a line of continuity to the text". This would facilitate the reading of the journal. The methodology and discussion of the results are presented in a clear and objective manner. However, the tables presented need to be improved. The format presented is not good.

The conclusion needs to be rewritten and greatly improved. The conclusion should be the corollary of the work. The authors should start by reminding the reader at the beginning of the conclusion of the aim of the work. Subsequently, they would present the results based on the proposed objectives. Do not forget to mention the importance of this study for society and for the formation of housing policies.

The bibliography is extensive and up-to-date, presenting very recent literature. However, attention should be paid to the formatting of bibliographic references. Some bibliographic references do not have a DOI.

Other issues related to the work presented:

1.   What is the main question addressed by the research? Is it relevant and interesting?

The issue of hedonic prices in the real estate market is very important to be studied because the real estate market has a strong impact of positive and negative externalities (positive and negative amenities). It is an important and relevant topic for the globalized society we live in. The biggest investment in most people's lives is in housing for their residence. The more literacy and knowledge about housing, the better it will be for society.

2.     How original is the topic? What does it add to the subject area compared with other published material?

According to the authors mentioned in the text, the topic is original and relevant to the country where this study is carried out, with no investigation in this area of research being known. This is an important work in this research area, as it also allows comparison with other countries.

3.     Is the paper well written? Is the text clear and easy to read?

Yes, the text is well written, clearly, objectively, and precisely. In this item, the paper is fine. However, in the discussion of results, they should confront with the literature review, namely where the results are coincident and where it deviates from the previous studies referred to in the literature review.

4.     Are the conclusions consistent with the evidence and arguments presented? Do they address the main question posed?
In the discussion of results, the work can be improved, confronting the results with the literature review, pointing out the similarities and differences with the studies referenced in the literature review. The conclusions must be improved.

5.     Is there any current bibliography that can be suggested, with the aim of improving the paper?

The bibliography is extensive and current, featuring very recent bibliography. Therefore, the bibliography needs to be improved. Attention should be paid to the formatting of bibliographic references. Some bibliographic references do not show DOI.

Author Response

Reviewer 4

Comments and Suggestions for Authors

Proximity to neighbourhood services and property values in urban area: An evaluation through Hedonic Pricing Model

Comments and Suggestions for Editor and Authors

The use of hedonic price models is an unavoidable topic of research on prices in the real estate market. Although there are international studies on the use of hedonic models, the authors refer that in their country it is a pioneering work, so it will be an important work for that region of the globe. In addition, it is very important for the academic community to develop research on a current topic, such as this case. The work is well-structured. The abstract, introduction, and literature review are written carefully and balanced. The aim of the work could (should) be mentioned in the introduction and at the beginning of the conclusion, thus giving "a line of continuity to the text". This would facilitate the reading of the journal. The methodology and discussion of the results are presented in a clear and objective manner. However, the tables presented need to be improved. The format presented is not good.

We re-define the objectives in “introduction section” and improve the tables

The conclusion needs to be rewritten and greatly improved. The conclusion should be the corollary of the work. The authors should start by reminding the reader at the beginning of the conclusion of the aim of the work. Subsequently, they would present the results based on the proposed objectives. Do not forget to mention the importance of this study for society and for the formation of housing policies.

Thank you for your comments; we rewrite the discussion and conclusion section once again with proper structure.

The bibliography is extensive and up-to-date, presenting very recent literature. However, attention should be paid to the formatting of bibliographic references. Some bibliographic references do not have a DOI.

We try at level best to add all the relevant data related to bibliography. We also add more relevant/precise papers associated with housing market and hedonic modeling.

Other issues related to the work presented:

  1. What is the main question addressed by the research? Is it relevant and interesting?

The issue of hedonic prices in the real estate market is very important to be studied because the real estate market has a strong impact of positive and negative externalities (positive and negative amenities). It is an important and relevant topic for the globalized society we live in. The biggest investment in most people's lives is in housing for their residence. The more literacy and knowledge about housing, the better it will be for society.

Thank you very much for our pervious time to read our paper and valuable comments to improve the study for society well-being and urban planning and  policies making.

  1. How original is the topic? What does it add to the subject area compared with other published material?

According to the authors mentioned in the text, the topic is original and relevant to the country where this study is carried out, with no investigation in this area of research being known. This is an important work in this research area, as it also allows comparison with other countries.

Yes this study making 1st attempt in this area of Gujrat City in Pakistan, thanks for you feedback.

  1. Is the paper well written? Is the text clear and easy to read?

Yes, the text is well written, clearly, objectively, and precisely. In this item, the paper is fine. However, in the discussion of results, they should confront with the literature review, namely where the results are coincident and where it deviates from the previous studies referred to in the literature review.

We rewrite the discussion and link with past literature

  1. Are the conclusions consistent with the evidence and arguments presented? Do they address the main question posed?
    In the discussion of results, the work can be improved, confronting the results with the literature review, pointing out the similarities and differences with the studies referenced in the literature review. The conclusions must be improved.

 We rewrite the conclusions

  1. Is there any current bibliography that can be suggested, with the aim of improving the paper?

The bibliography is extensive and current, featuring very recent bibliography. Therefore, the bibliography needs to be improved. Attention should be paid to the formatting of bibliographic references. Some bibliographic references do not show DOI.

We try to add all of available information /data related to bibliography, hope its work

Reviewer 5 Report

In the manuscript “Proximity to neighbourhood services and property values in urban area: An evaluation through Hedonic Pricing Model” the authors used Geographical Information Science through digitizing the point of interest in study area for spatial modeling of data collection points and Multi Linear Regression as a statistical analysis of considered data. This manuscript is well organized, and the drawn conclusions are coherent with the obtained results. The references should be updated to include more recent studies. 

Line 38: The keywords should be arranged alphabetically.

Lines 78 – 79: I think that you should add this recent and interesting reference to support your sentence: “The major change in land use around the world is expansion of built up environment”. I would like to suggest:

Fraissinet, M., et al. (2023). Responses of avian assemblages to spatiotemporal landscape dynamics in urban ecosystems. Landscape Ecology, 38(1), 293-305.

Lines 143 – 145: The authors should better highlight their hypothesis and predictions.

Lines 414 – 442: The authors should discuss better their results. The discussion sections is inexistent. This part of the manuscript should be expanded.

Lines 442: I think that you should add this recent and interesting reference as an example to support your sentence: “...values in this area through GIS and statistical measurements”. I would like to suggest:

Ancillotto, L., et al. (2019). The importance of ponds for the conservation of bats in urban landscapes. Landscape and Urban Planning, 190, 103607.

Author Response

Reviewer 5

Comments and Suggestions for Authors

In the manuscript “Proximity to neighbourhood services and property values in urban area: An evaluation through Hedonic Pricing Model” the authors used Geographical Information Science through digitizing the point of interest in study area for spatial modeling of data collection points and Multi Linear Regression as a statistical analysis of considered data. This manuscript is well organized, and the drawn conclusions are coherent with the obtained results. The references should be updated to include more recent studies. 

More references has been added

Line 38: The keywords should be arranged alphabetically.

Now they are arranged alphabetically.

Lines 78 – 79: I think that you should add this recent and interesting reference to support your sentence: “The major change in land use around the world is expansion of built up environment”. I would like to suggest:

Fraissinet, M., et al. (2023). Responses of avian assemblages to spatiotemporal landscape dynamics in urban ecosystems. Landscape Ecology, 38(1), 293-305.

Thank you for your comment, the mentioned references is cited/linked with this sentence.

Lines 143 – 145: The authors should better highlight their hypothesis and predictions.

The study has been redefine, we formulate the hypothesis and predications in the 1st section of this study

Lines 414 – 442: The authors should discuss better their results. The discussion sections is inexistent. This part of the manuscript should be expanded.

This whole of the section is rewrite

Lines 442: I think that you should add this recent and interesting reference as an example to support your sentence: “...values in this area through GIS and statistical measurements”. I would like to suggest:

Ancillotto, L., et al. (2019). The importance of ponds for the conservation of bats in urban landscapes. Landscape and Urban Planning, 190, 103607.

This reference has been added in this line for support of this sentence.

Round 2

Reviewer 4 Report

Dear Editor and authors,

After reviewing the Paper - Proximity to neighborhood services and property values in urban area: An evaluation through Hedonic Pricing Model - again, I understand that it has substantially improved. However, I noticed that there are some DOIs that exist but were not added to the bibliography, for example: Lu, X. H., & Ke, S. G. (2018). Evaluating the effectiveness of sustainable urban land use in China from the perspective of sustainable urbanization. Habitat International, 77, 90-98. - The DOI is: https://doi.org/10.1016/j.habitatint.2017.10.007. You may consider adding it when you edit the paper. Nonetheless, I am pleased with the improvements made. Thank you and good luck.

Author Response

Dear Reviewer, 

Thank you very much for your valuable comments to improve our paper, please note the mentioned reference DOI is added in reference list and paper body. you can check it (highlight in Green color). 

Reviewer 5 Report

Well done!

Author Response

Dear Reviewer,

Thank you for your comments, the mentioned references has been added with DOI at number 12, ( green line in color in paper body), 
